# Brain Signal Generation and Data Augmentation with a Single-Step Diffusion Probabilistic Model

## Abstract

Brain-computer interfaces based on deep learning rely on large amounts of high-quality data. Finding publicly available brain signal datasets that meet all requirements is a challenge. However, brain signals synthesized with generative models may provide a solution to this problem. Our work builds on diffusion probabilistic models (DPMs) and aims to generate brain signals that have the properties needed to develop further classification models based on deep learning. We show that our DPM can generate high-quality event-related potentials (ERPs) and motor imagery (MI) signals. Furthermore, with the progressive distillation of the model, subject-specific data can be produced in a one-step reverse process. We augment publicly available datasets and demonstrate the impact of the generated signals on a deep learning classification model. DPMs are versatile models, and this work shows that brain signal processing is one of many other tasks in which these models can be useful.

## 1 Introduction

Electroencephalography (EEG) is undoubtedly one of the most popular brain mapping technologies, which is widely used in research and clinical diagnosis (de Aguiar Neto & Rosa (2019), van Mierlo et al. (2020), Wang et al. (2020)). EEG records the neural activity of the brain in a non-invasive manner. (Biasiucci et al. (2019)) EEG is less complex and cheaper than other brain imaging technologies. EEG has one of the best temporal resolutions. However, the spatial resolution of the technology is quite poor due to its heavy dependence on the number of electrodes used for signal recording and non-invasiveness. (Craik et al. (2019b))

Brain-computer interfaces (BCIs) connect the brain and external processing devices, making it possible to perform tasks using only brain signals. BCIs can help in everyday life for people with limited movement and communication abilities (Pandarinath et al. (2017)). BCIs are also applied in many other fields from healthcare (Galán et al. (2008), Vilela & Hochberg (2020)) to entertainment (Finke et al. (2009)). BCIs are often based on EEG due to the ability of the technology to measure signals with only a couple of milliseconds difference and its relatively low cost and more comfort. The measurements are then processed by a decoder unit in the BCI that turns the recorded temporal and frequency patterns into actions. (Lotte et al. (2018))

In recent years, deep learning (DL) algorithms have become more and more commonly used in EEG signal processing (Roy et al. (2019), Craik et al. (2019a), Kotowski et al. (2020)). DL models can decode brain signals with high accuracy. However, developing DL models requires a large amount of high-quality data. The size and quality of publicly available data sets are limited, also often insufficient and imbalanced. Recording a new data set can be highly resource-consuming and requires professionals to check the measurements. Another option to augment data sets is data synthesis. (Lashgari et al. (2020))

Score-based models (Tashiro et al. (2021), Song et al. (2021)), diffusion probabilistic models (DPMs) (Ho et al. (2020), Luo & Hu (2021)) and generative adversarial networks (GANs) (Liu et al. (2021), Chan et al. (2021)) hold the state-of-the-art in deep-learning-based generative modelling. The recent advances show the performance and effectiveness DPMs over GANs in both image (Dhariwal & Nichol (2021)) and audio generation (Kong et al. (2021)). There are a handful

of works for brain signal generation with GANs (Xu et al. (2022), Hartmann et al. (2018), Fahimi et al. (2019), Panwar et al. (2020)). To the best of our knowledge, there are no works examining the capabilities of DPMs or score-based models in multi-channel EEG signal generation tasks.

The structure of this work is as follows: in Section 2 we present the background on the DPM framework that we used in this paper, followed by a brief description of the progressive distillation process in Section 3. Our EEGWave architecture is presented in Section 4. The description of the experiments with the used datasets and procedures are given in Section 5. Finally, we conclude our work and thoughts in Section 6.

## 2 CONTINUOUS-TIME DIFFUSION MODELS

The distribution of the training data set is given as $p(x)$. Let $x \in \mathbb{R}^{E \times L}$, where E is the number of electrodes (or EEG channels) and L is the length of the recorded sequence. In a continuous-time diffusion framework (Kingma et al. (2021)), in the forward and reverse diffusion processes, there are latent variables that are denoted by $z_t$. For every time step, where $t \in [0, 1]$, the latent variables have the same shape as the training data samples ($z_t \in \mathbb{R}^{E \times L}$).

The **forward diffusion process**, which is a Gaussian process in continuous time can be given as:

$$q(z_t|x) = \mathcal{N}(z_t; \alpha_t x, \sigma_t^2 \mathbf{I}) \tag{1}$$

,where $\alpha_t$ and $\sigma_t^2$ are smooth, differentiable, positive scalar-valued functions. With $\alpha_t$ and $\sigma_t^2$ the log signal-to-noise ratio is given as: $\lambda_t = \log(\alpha_t^2/\sigma_t^2)$, which decreases strictly monotonically as $t \to 1$. For any $0 \leq s \leq t \leq 1$, the following Gaussian conditional distribution can be given:

$$q(z_t|z_s) = \mathcal{N}(z_t; \frac{\alpha_t}{\alpha_s} z_s, \tilde{\sigma}_{t|s}^2 \mathbf{I}), \quad \tilde{\sigma}_{t|s}^2 = (1 - e^{\lambda_t - \lambda_s})\sigma_t^2 \tag{2}$$

The **reverse process** is based on the a posteriori distribution:

$$q(z_s|z_t, x) = \mathcal{N}(z_s; \tilde{\mu}_{s|t}(z_t, x), \tilde{\sigma}_{s|t}^2 \mathbf{I}) \tag{3}$$

, where $s \leq t$ and

$$\tilde{\mu}_{s|t}(z_t, x) = e^{\lambda_t - \lambda_s} \frac{\alpha_s}{\alpha_t} z_t + (1 - e^{\lambda_t - \lambda_s})\alpha_s x, \quad \tilde{\sigma}_{s|t}^2 = (1 - e^{\lambda_t - \lambda_s})\sigma_s^2 \tag{4}$$

In this framework, $x$ in the reverse process is predicted by the neural network $\tilde{x}_\theta(z_t, \lambda_t)$ with the parameter set $\theta$.

Data **inference** is done by sampling a latent white noise variable $z_1$ at $t = 1$, setting a noise controlling $\gamma$ factor and iteratively applying the following, until $t = 0$ (Salimans & Ho (2022)):

$$z_s = \tilde{\mu}_{s|t}(z_t, \tilde{x}_\theta(z_t, \lambda_t)) + \sqrt{(\tilde{\sigma}_{s|t}^2)^{1-\gamma}(\tilde{\sigma}_{t|s}^2)^\gamma}\epsilon, \quad \epsilon \sim \mathcal{N}(0, \mathbf{I}) \tag{5}$$

During **training**, the model is aimed to maximize the variational lower bound (ELBO) on the log-likelihood of the data. However, with a re-parameterization, the weighted ELBO can be given as the weighted mean squared error objective:

$$\min_\theta L(\theta) = \mathbb{E}_{\epsilon,t}\Big[\omega(\lambda_t)\|x - \tilde{x}_\theta(z_t, \lambda_t)\|_2^2\Big] \tag{6}$$

, where $\omega(\lambda_t)$ weighting is choosable, however it is $\omega(\lambda_t) = \max(\frac{\alpha_t^2}{\sigma_t^2}, 1)$ in our approach. Salimans & Ho (2022)

## 3 PROGRESSIVE DISTILLATION

Diffusion models need many iterations during sampling to synthesize data, making them significantly slower than GANs. Recent studies (Luhman & Luhman (2021), Kong & Ping (2021)) presented multiple ways to fasten the inference from which we applied progressive distillation (Salimans & Ho (2022)) as we found this approach the most efficient one.

Progressive distillation is based on a teacher-student setup, where the student model approximates the teacher with halved sampling steps. At first, a teacher model is trained as a continuous-time diffusion model. Then a number of finite discrete sampling steps is given for the teacher and the student, denote it as $T_t$ and $T_s = T_t/2$. The distillation process is iterative, where each iteration starts with weight initialization of the student model with the teacher's parameters. The target $\tilde{x}$ is then calculated from the latent variable $z_t$ by sampling in 2 DDIM steps using the teacher model to get the predictions at time step $(t - 0.5)/T_s$ and the $(t - 1.)/T_s$. The student has to denoise the same $z_t$ in 1 reverse DDIM step to get the approximate result for $\tilde{x}$. When the student converges, $T_s$ is halved, the student becomes the teacher, and the process is repeated.

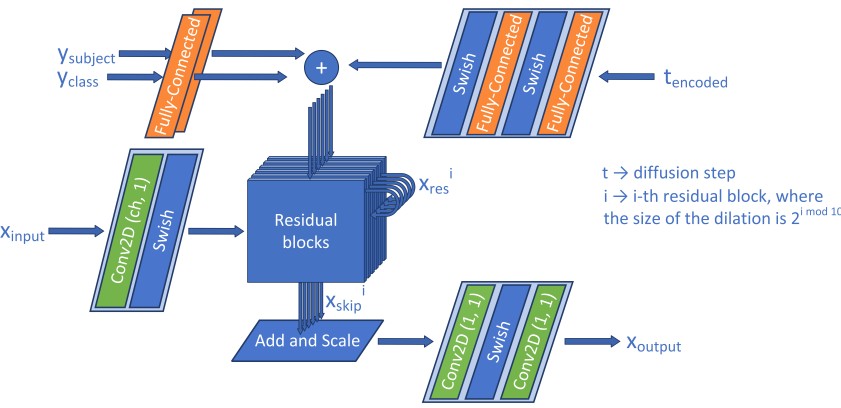

Figure 1: EEGWave architecture.

## 4 ARCHITECTURE

Our architecture aims to synthesize multi-channel EEG signals as many works on brain signal processing show the benefits of processing signals recorded on multiple electrodes. Therefore, our model is $\tilde{x}_\theta : \mathbb{R}^{E \times L} \times \mathbb{R} \to \mathbb{R}^{E \times L}$ that builds on non-causal bi-directional dilated convolutions.

Many works on multi-channel brain signal processing handle the measurements as 2-dimensional samples, similarly to images, to make use of spatio-temporal features. In GANs and VAEs, this approach means that EEG signals are synthesized through up-sampling interpolations or transposed convolutions. Data synthesis in this approach with time-series signals can be ill-posed due to the heavy influence of these operations' temporal and spectral artifacts that occur in the generated data.

Our approach builds on the work of Kong et al. (2021). We omit up- and down-sampling layers to avoid temporal and spectral artifacts in the synthesized data. We build on the residual layers of DiffWave. Although EEG epochs are not as long as audio samples, we keep bi-directional dilated convolutions with smaller dilations as they can help maintain global context and consistency through the epochs. We realize the architecture with 2-dimensional convolutions. Our input convolution layer maps the multi-channel data into a single-channel by applying $(E, 1)$ sized filters, where E is the number of EEG channels. We use kernels with size $(1, K)$ in the dilated convolution layers as we process 1-dimensional data in these blocks. The output point-wise convolution layer produces the 1-dimensional signals with the number of EEG channels $(N \times E \times 1 \times L)$. Then this data is reshaped to match the dimensions of the input $(N \times 1 \times E \times L)$. (We implemented the architecture in PyTorch.)

$\lambda_t$ is embedded through two global and one residual-local linear layers and added to the residual features. The class and subject conditions are handled the same way, the only difference being that they are not encoded in any way. The architecture is shown in Figure 1.

In the experiments given in this work, we use a rather deep than wide neural network with 32 residual layers and 64 channels in each convolution layer. $\lambda_t$, one-hot encoded class, and subject conditions are embedded into 512 dimensions in the global linear layers. We use a kernel size of $(1, 3)$ in the dilated convolutions and a dilation cycle of $[1, 2, ..., 64]$ because empirically, we found it to be

beneficial to use a dilation cycle that with there are kernels that cover the whole data sample. $\lambda_t$ is embedded through two global and one residual-local linear layers and added to the residual features. The class and subject conditions are handled the same way, the only difference being that they are not encoded in any way.

# 5 EXPERIMENTS

## 5.1 DATASETS

**VEPESS dataset:**  the set contains visually evoked potentials from 18 subjects recorded by the authors of Robbins et al. (2018). The measurements were done by following the oddball paradigm. Subjects were presented with sequences of images that consisted of target and non-target types. Based on the type of the shown image, the subjects had to push the corresponding buttons. Data were recorded with an EEG device with 64 + 6 electrodes in the 10-20 standard configuration and sampled at 512 Hz. In this work, we use raw measurements. Signals were band-pass filtered from 1 - 40 Hz with a zero-phase filter, and epochs were extracted between $[0, 1]$ seconds from the onset of the target/non-target image. The epochs were normalized by channel-wise mean subtraction and deviation division into the range $[-1., 1.]$.

**BCI Competition IV Dataset IIa (BCIC4D2a):**  the set contains motor imagery signals from 9 participants. The subjects were asked to imagine the movement of their left and right hand, also their feet and tongue, for a couple of seconds after the instruction cue were presented on their screen. Data was recorded from 22 EEG and 3 EOG channels following the 10-20 standard system. The measurements were sampled at 250 Hz and band-pass filtered from 0.5 - 100 Hz. Furthermore, a notch filter at 50 Hz was applied to eliminate the line noise. We further band-pass filtered the signals between 4 - 38 Hz with a zero-phase filter and down-sampled them to 128 Hz. Following the work of , we extracted epochs from the recordings between $[0.5, 4]$ seconds from the onset of the cue and normalized them by channel-wise mean subtraction and deviation division. We excluded samples marked as rejected due to artifacts by the publishers of the set.

## 5.2 CLASS-CONDITIONAL SIGNAL GENERATION

The current experiment aims to generate EEG data of good quality from samples from a simple Gaussian noise distribution. We condition the generation process based on classes in the data sets. We use the VEPESS and the BCIC4D2a data sets to examine whether the model can capture the features of the sets that contain different types of signals. For both data sets, EEGWave has to model characteristics in both the time and frequency domain, but in a slightly different way. The signals from the VEPESS set are characterized mainly by their amplitude deviation and the corresponding latency in the time domain. On the other hand, the essential features of the samples from the BCIC4D2a set are in the frequency domain, specifically in the theta, alpha, and beta bands. Although the effect of progressive distillation on the quality of the generated data is examined in the following subsection, we also present the results from the distilled models, which generated the signals in a single step.

We chose only the RWGAN (Panwar et al. (2020)) as a baseline model because other deep learning models in the literature were either incompletely documented, designed only for single-channel EEG generation, or did not work based on the given information in the published work. We trained RWGAN exactly as the publishers did in their work.

We give both qualitative and quantitative results. The Inception Score (IS) (Salimans et al. (2016)), Frechet Inception Distance (FID) (Heusel et al. (2017)), spatial FID (Nash et al. (2021)), Precision, Recall (Kynkäänniemi et al. (2019)) scores are computed based on EEGNet (Lawhern et al. (2018)), which was trained on the data sets separately the same way as the authors of EEGNet did. We also use Sliced Wasserstein Distance (SWD) (Wu et al. (2019)) and Gaussian Mixture Model (GMM) differences (Panwar et al. (2020)) to measure the differences between real and generated distributions directly. We also found that qualitative results are a good way to have a greater understanding and a deeper interpretation of results.

Table 1: The quality of the generated EEG signals on the VEPESS set was measured indirectly (with the feature maps of EEGNet) and directly (through distribution measures).

| Origin | IS ↑ | FID ↓ | sFID ↓ | Prec ↑ | Rec ↑ | SWD ↓ | dGMM ↓ |
|---|---|---|---|---|---|---|---|
| Train set | 1.2522 | 0 | 0 | 1.0 | 1.0 | 0 | 0 |
| Test set | 1.2360 | 0.0315 | 1.3264 | 0.9199 | 0.9369 | 0.9851 | 49.8316 |
| RWGAN | 1.1761 | 2.2708 | 3.0530 | 0.8865 | **0.9087** | 1.1769 | 119.0032 |
| EEGWave | **1.2284** | **0.1479** | **1.5266** | **0.9238** | 0.8740 | **1.0915** | **81.1503** |
| EEGWave 1x | 1.2146 | 1.4212 | 5.6885 | 0.8063 | 0.8284 | 1.7785 | 141.4728 |

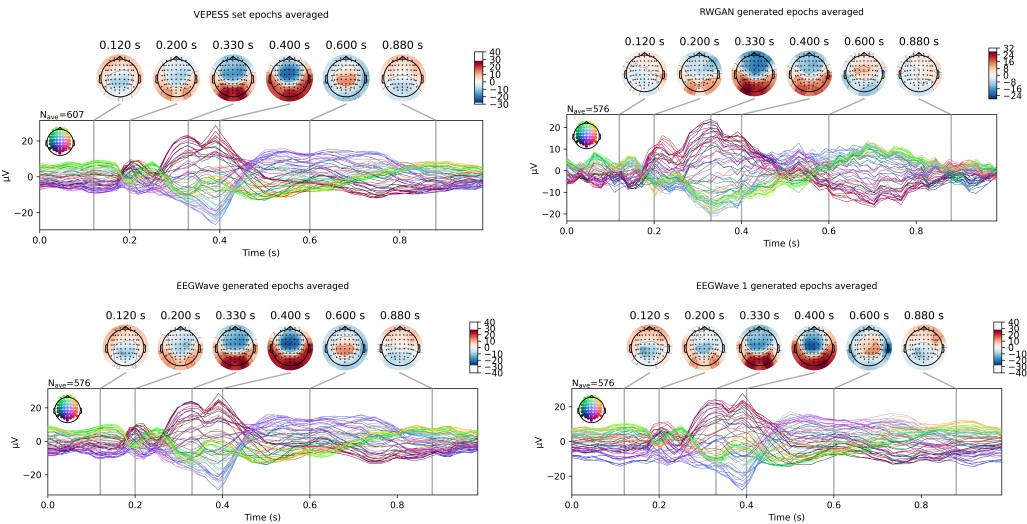

Figure 2: Comparison of averaged real and generated samples by different models on the VEPESS set. The averaging was done across the epochs, and the results are plotted on all 64 channels.

**VEPESS results:** first, we present the results on the VEPESS set. The quantitative metrics are given in Table 1. We also give scores measured on the test set to understand the generated signals' results better. Figure 2 presents the averaged target class samples from the original and the generated sets. Based on the given results, it can be said that EEGWave captured the ERPs' main features in the VEPESS set. RWGAN also generated good signals, although these samples are heavily contaminated by frequency artifacts, which, in our hypothesis, are mostly the results of the up-sampling layers (also mentioned frequently in the literature). The single-step EEGWave has slightly poorer scores in the table, but the generated signals seem to have more fidelity than the ones from the RWGAN.

**BCIC4D2a results:** for the BCIC4D2a set, the quantitative results are given in Table 2. Figure 3 presents the power spectral densities distributed over the scalp in the theta, alpha, and beta frequency bands for the left-hand class. Generated signals from the EEGWave 1024 step, EEGWave 1 step, and RW-

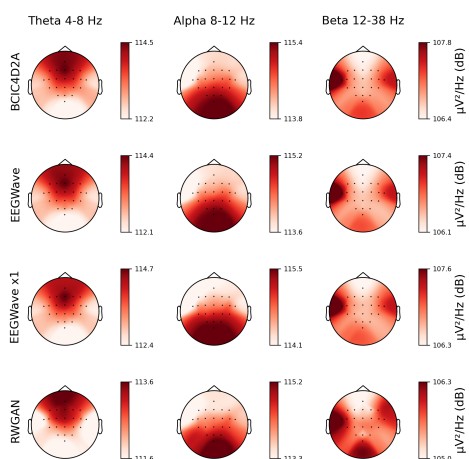

Figure 3: Real and generated signal PSD top plots from the BCIC4D2a set.

Table 2: The quality of the generated EEG signals on the BCIC4D2a set was measured indirectly (with the feature maps of EEGNet) and directly (through distribution measures).

| Origin | IS ↑ | FID ↓ | sFID ↓ | Prec ↑ | Rec ↑ | SWD ↓ | dGMM ↓ |
|---|---|---|---|---|---|---|---|
| Train set | 1.3998 | 0 | 0 | 1.0 | 1.0 | 0 | 0 |
| Test set | 1.3594 | 0.1798 | 80.4392 | 0.9040 | 0.9172 | 0.4781 | 2.4963 |
| RWGAN | **1.2949** | 7.2315 | 142.8659 | 0.4740 | 0.4123 | 1.6983 | 751.2371 |
| EEGWave | 1.1881 | **3.1708** | 51.9159 | **0.9668** | **0.5724** | 1.3566 | 597.6406 |
| EEGWave 1x | 1.2136 | 3.3100 | **42.8244** | 0.9282 | 0.5245 | **1.0874** | **430.7368** |

GAN models are compared to the real signals from the dataset. The results show that the single-step EEGWave model could perform slightly better than RWGAN in this experiment. The top plots show that all models were able to learn the main frequency features. The two EEGWave models outperformed the GAN. The figures corresponding to the rest of the classes are given in Appendix A.1.

## 5.3 SUBJECT SPECIFICITY

Brain signals vary not just between classes but subjects. In many cases, it could be beneficial to generate data for only specific subjects, e.g., imbalanced sets, fine-tuning. We use the VEPESS set to show that subject-specific features can be learned and reproduced by our model. Subject information is one-hot encoded and injected into the network similarly to the signal class labels. The model is trained the same way as in the class-conditional case. During inference, the signal generation process is conditioned on the class labels and the subjects. We then compare the generated signals from each subject to the real signals from all subjects to examine whether the model could learn each subject's features. The generated signals are visualized in top plots. Furthermore, we measure the SWD and sFID metrics between the subject-specific distributions. For the sFID calculation, the same EEGNet model is used as in the previous experiment.

**Results:** the metrics in Figure 4 imply that the distributions of the real and generated signal corresponding to the same subjects are closer to each other than to other subject data. Figure 5 visually supports this implication. The averaged ERP epochs of each subject are visually easily distinguishable from each other. Although we only present a few examples here, we include the rest of the top plots in Appendix A.2.

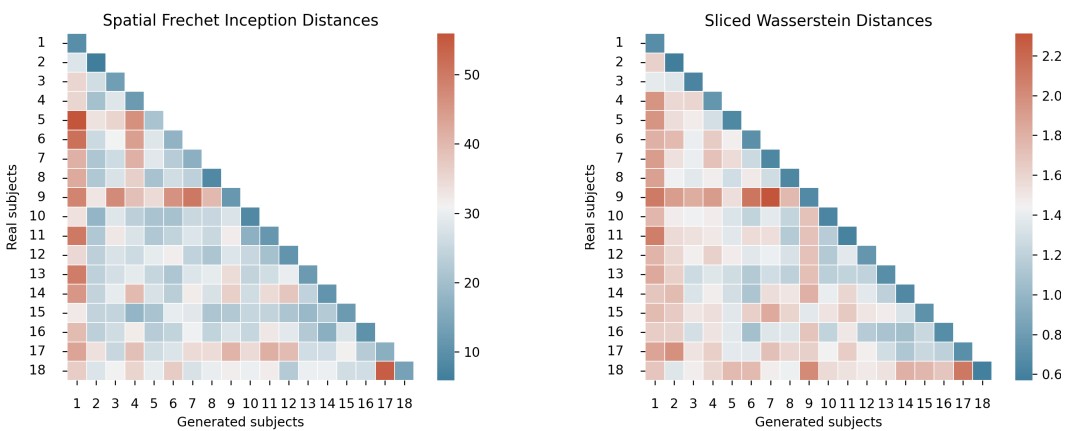

Figure 4: Two types of distance metrics were measured between the distributions of the real and generated subjects from the VEPESS set to examine the ability of the model to learn subject-specific features. (The lower, the better.)

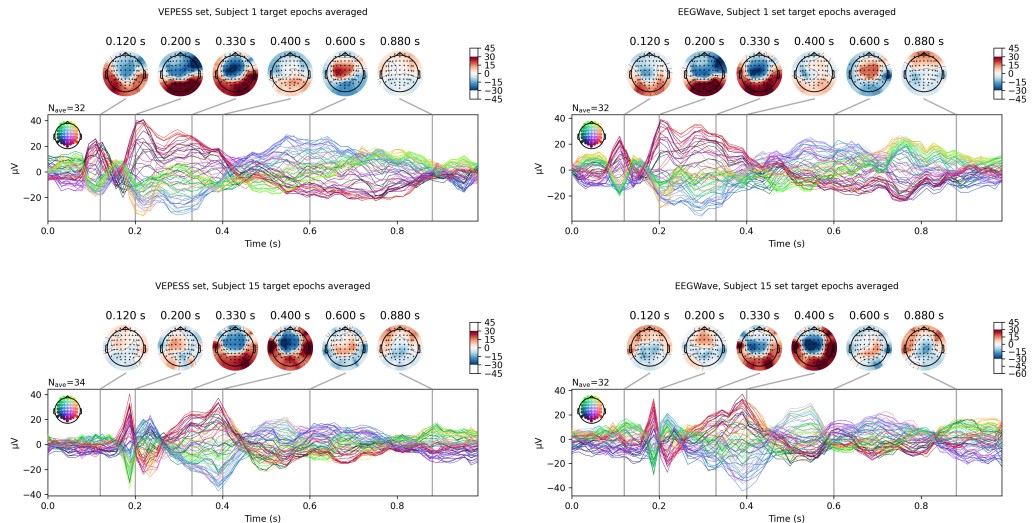

Figure 5: Comparison of the averaged real and generated ERP epochs from the VEPESS set for Subjects 1 and 15.

## 5.4 AUGMENTATION

In the augmentation task, we try to improve the performance of EEGNet on the BCI4D2a dataset. A generated dataset is created with the single-step EEGWave model with the same amount of signals as the original set. The original dataset is split into train, validation, and test subsets with $0.7\%, 0.15\%, 0.15\%$ ratios. The subsets are created to contain the same number of signals from each subject. We try to improve the accuracy of the EEGNet model with two approaches:

1. we double the size of the training subset by mixing the same amount of generated signals as the number in the original subset (6)

2. pre-train EEGNet on the generated data and then re-train this initialized model with the real signals (7)

In both cases, we only stop the training if over-fitting is detected via monitoring the validation loss. We use Adam optimizer with a learning rate of $1e - 3$.

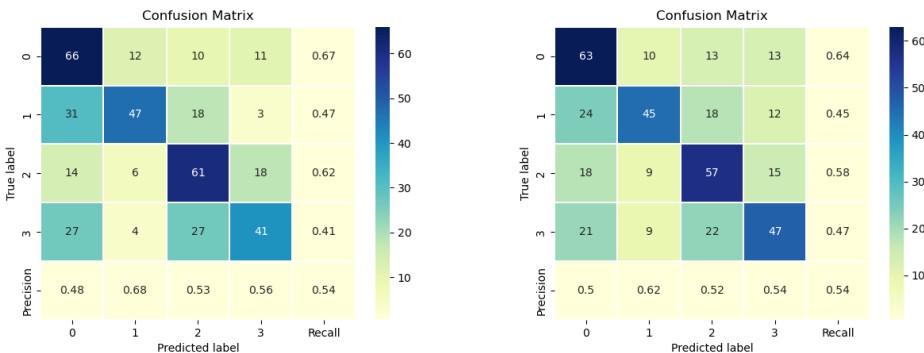

Figure 6: The effect of training EEGNet with the real-generated mixed training subset. The left confusion matrix shows the test results without augmentation, while the right one shows the training results with the mixed-set augmentation.

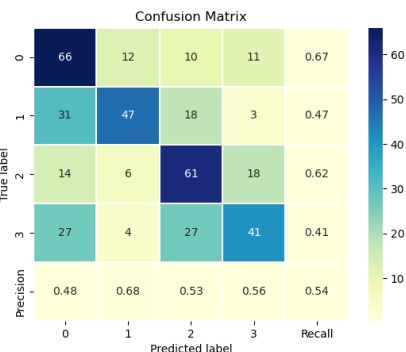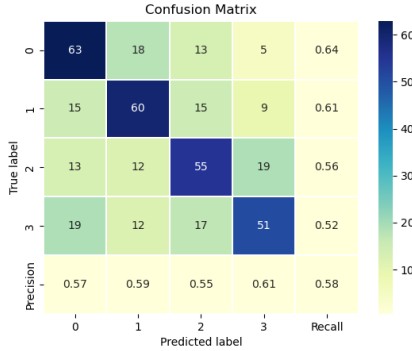

Figure 7: The effect of pre-training EEGNet with the generated signals. The left confusion matrix shows the test results without pre-training, while the right one shows the results with pre-training.

**Results:** although the mixed training gave slightly better accuracy on the validation subset during training, it did not improve the accuracy of the test set. On the contrary, with the pre-training approach, the model converged faster and achieved better accuracy on the test. These observations imply that although EEGWave was able to learn the main characteristics of the real signals, it could not produce signals that are diverse enough to regularize EEGNet.

## 5.5 DISTILLATION

By progressive distillation, we aim to attain a DPM model that can synthesize EEG signals in a single step. We distill EEGWave on the VEPESS and BCIC4D2a data sets to examine the effect of the process. The distillation is started at 1024 steps, and we continue it until the single-step generation is reached. The number of training iterations in steps 2 and 1 is doubled, following the work of _. The distilled models with different inference steps are saved and compared to the initial model generating signals through the same number of steps as the distilled ones. We evaluate the generated signals at these steps and present results in Figures 8 and 9.

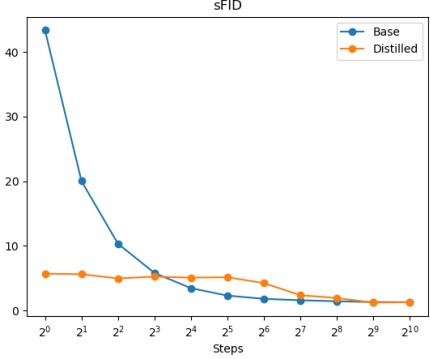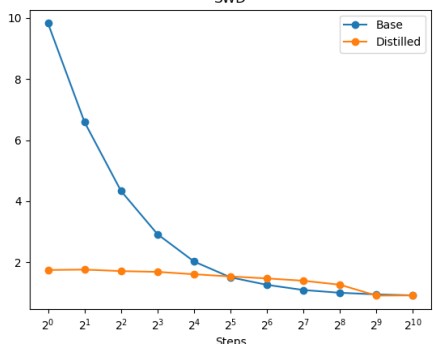

Figure 8: Effect of the distillation of EEGWave trained on the VEPESS data set. F-Score is calculated from the Precision and Recall scores with $\beta = 2$.

**Results:** while at a higher number of steps ($> 8$), distillation did not result in a better-performing model. At steps $\leq 8$, the metrics show much better models than the ones without distillation. In the case of the VEPESS set, the scores achieved with the single-step distilled model are much closer to the scores of the initial model than without distillation. This shows that distillation can be a good option for achieving a DPM that can effectively balance the trade-off between fast sampling and high-quality samples. Interestingly, in the case of the BCIC4D2a set, the scores of the distilled mod-

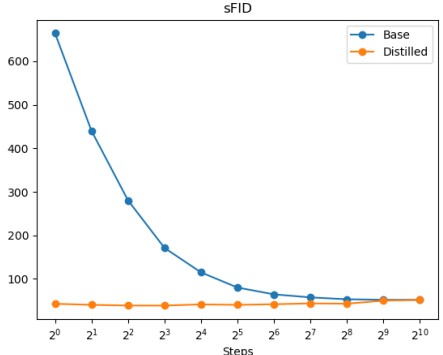 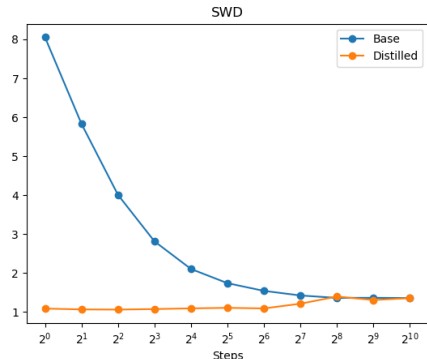

Figure 9: Effect of the distillation of EEGWave trained on the BCIC4D2a data set. F-Score is calculated from the Precision and Recall scores with $\beta = 2$.

els show better performance than those of the initial model. This implies that the distilled models are more likely to generate EEG signals that are copies of the ones from the train set, resulting in better metrics due to higher fidelity but a narrower learned distribution.

## 6 CONCLUSION

Our work shows a novel way of generating brain signals that can be useful in augmentation tasks. Although this work aimed to examine as many aspects of brain signal generation as possible, there is much yet to explore. We believe the current work shows that DPM-based brain signal generation is a very feasible task and can be used to create datasets that help improve deep learning models in classification tasks.

This work is mainly limited in that the quality and diversity of EEG signals can not be measured the same way as in the case of images. The metrics commonly used in image synthesis tasks often give contrary results in brain signal generation.

ERP and MI signals were generated conditioned on class labels. The performance of our single-step DPM was close to 1024-step DPM and the RWGAN. We also showed that EEGWave could learn subject-specific features. In the augmentation task, the generated signals were the most useful for the pre-training of EEGNet, before re-training on the original set. The distillation results show that progressive distillation is an excellent approach to obtaining a DPM with a low number of inference steps that can generate signals of good quality. The diversity of the generated signals is still an open question, as well as the metrics that can measure the realness of the generated data. We hope that DPM-based signal generation will be much more explored in the future, and we are eager to see the development of this field.

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

# A APPENDIX

## A.1 GENERATED MOTOR IMAGERY SIGNALS

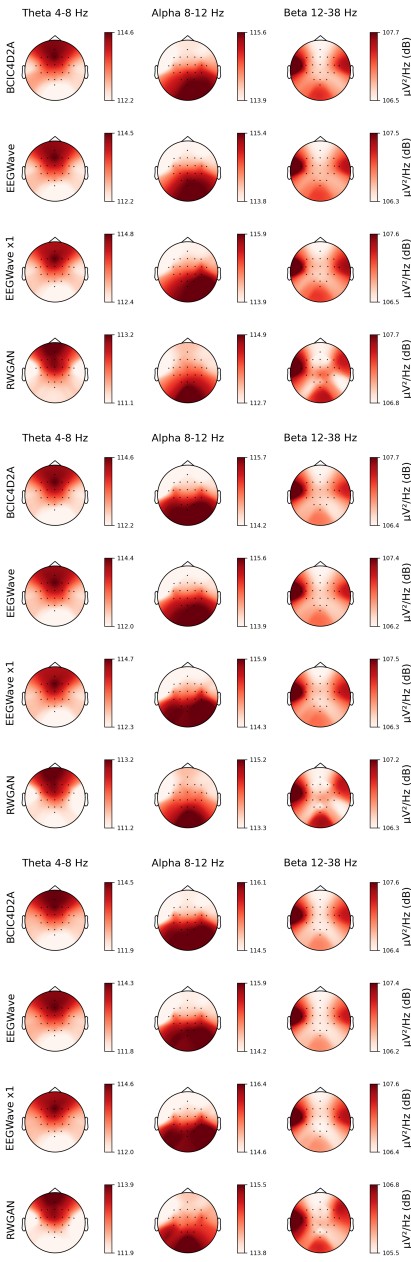

Figure 10: PSD top plots of the generated signals from the BCIC4D2a dataset. The top, middle, and bottom figures visualize the right hand, feet, and tongue classes.

## A.2 GENERATED SUBJECT-SPECIFIC ERP SIGNALS

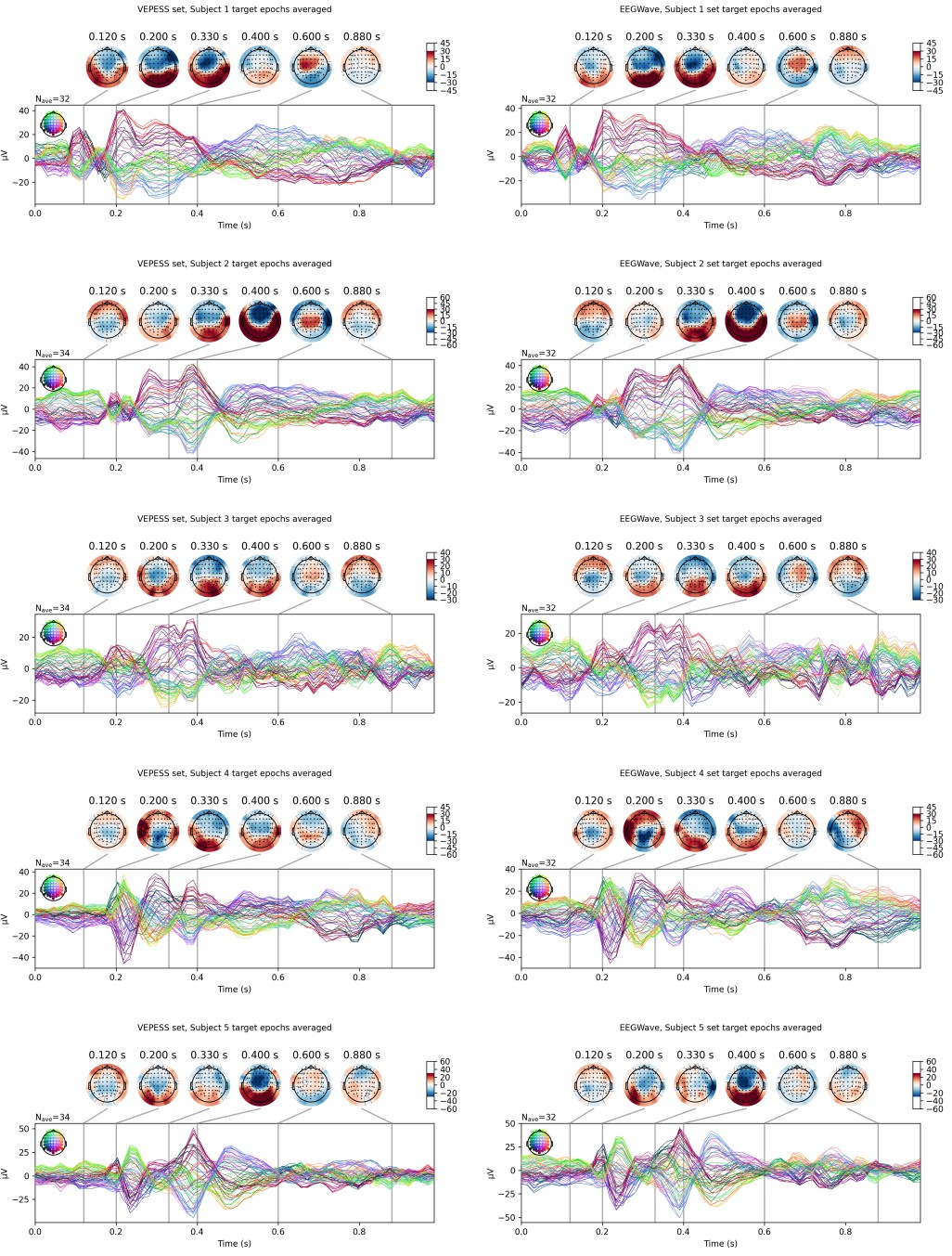

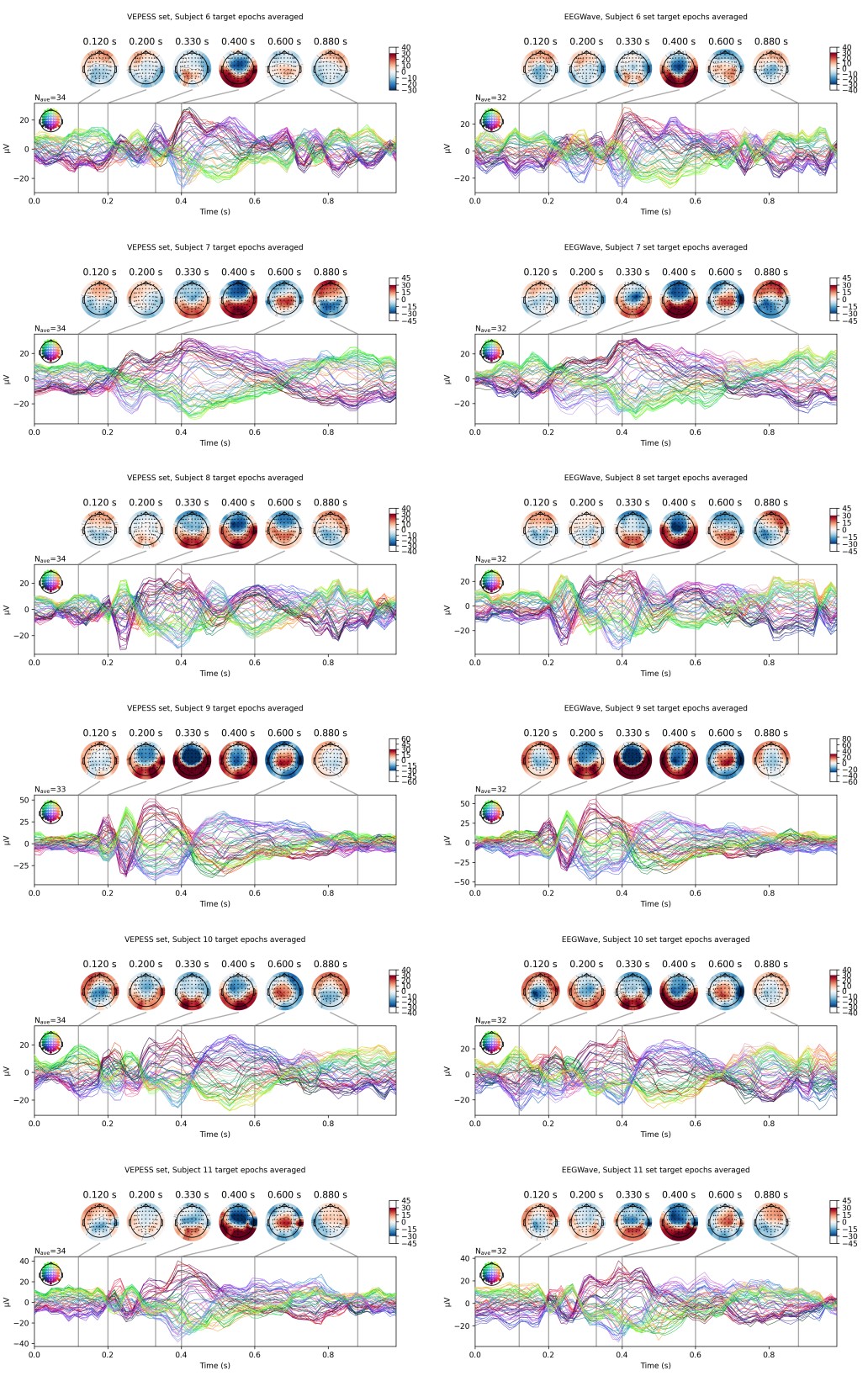

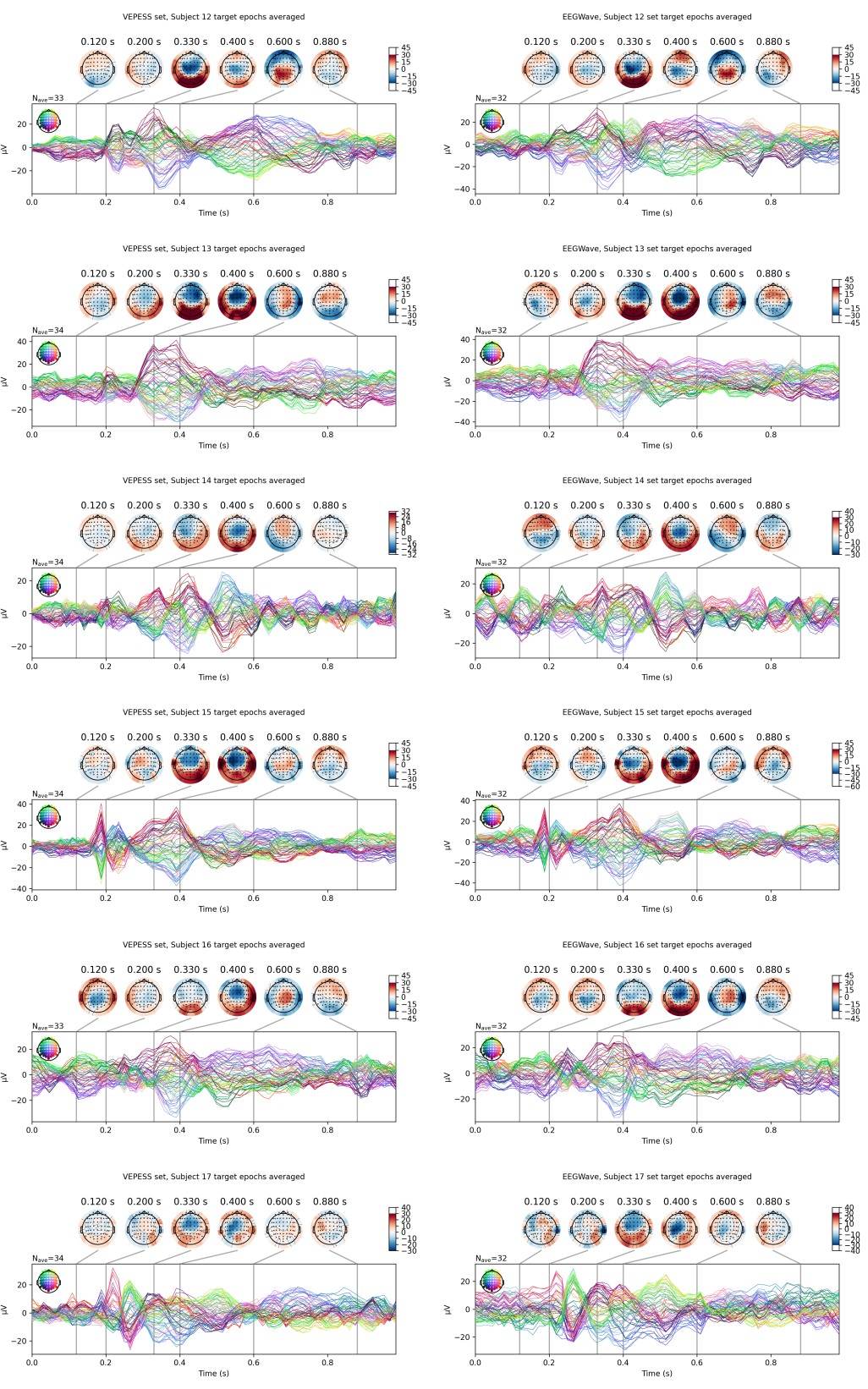

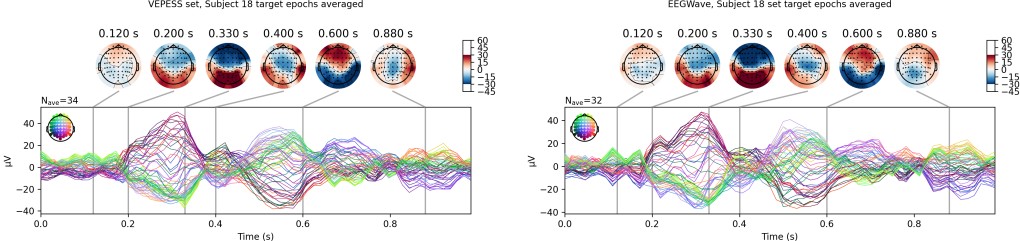

