# OpenReview forum: "Brain Signal Generation and Data Augmentation with a Single-Step Diffusion Probabilistic Model"
_ICLR.cc/2023/Conference — Submitted to ICLR 2023_

### Official Review · Reviewer_UL6z · 2022-10-20

**Confidence:** 4
**Correctness:** 2
**Technical Novelty And Significance:** 2
**Empirical Novelty And Significance:** 2
**Recommendation:** 1

**Clarity, Quality, Novelty And Reproducibility:**

The clarity has a large room for improvement.
The quality and the novelty are both not sufficient.


**Strength And Weaknesses:**

Strength

1. The paper presented a set of interesting application tasks/experiments on EEG datasets.
2. Many tasks are considered, including class-conditioned generation, subject-conditioned generation, and classification.

Weaknesses
1. Major concern 1: the method lacks novelty
- The presented work is a simple application of existing diffusion model + progressive distillation on EEG datasets. No novel method is presented.

2. Major concern 2: experimental evaluation
- There exists only one benchmark model throughout the paper. The authors give their reason as “We chose only the RWGAN (Panwar et al. (2020)) as a baseline model because other deep learning models in the literature were either incompletely documented, designed only for single-channel EEG generation, or did not work based on the given information in the published work.” However, I found it hardly a valid reason for not implementing enough benchmarks. (1) If the model isn’t completely documented, one should try to implement it based on customized design and model choice that is fairly selected. (2) One could always try to implement classic models such as naive VAE or GAN based on the presented setting. (3) “The model does not work” also is an unvalidatable claim as it is unclear if it is reviewer’s coding issue or if it is the models’ issue.
- Due to the lack of benchmarking models, both Table 1 and Table 2 are not really informative and do not tell if the model perform good enough. First of all, for many scores the test-set score is better than the model’s performance, which only give a controversial conclusion because it either indicates that (1) the dataset is distributed in a certain diverse way and the model does not overfit (which indicates in certain case the lower the score the better), (2) the model does not perform well enough to mimic the training set structure. That is why most existing works under this setting would just train many models on the whole dataset and compare different models’ performance.
- The augmentation experiments is not complete. There are no benchmarking models. One thing that can be done is to use the generated data as augmentation and compare it with traditional augmentation (e.g. noise). Also one could benchmark the generated data from this model to the generated images from other models (the GAN that is implemented beforehand).
- Overall, there isn’t sufficient quantitative evaluation as well.

3. The writing isn’t clear.
- There are many grammar errors and wording issues throughout.

4. Related works are not summarized and compared properly.


**Summary Of The Paper:**

This work attempts to apply diffusion models to synthesize EEG data as data augmentation. Specifically, the work uses the continuous-time diffusion framework with progressive distillation as their methodology. The synthesized data is validated based on training on the VEPESS dataset and the BCIC4D2a dataset, where a set of model validation tasks are performed including (1) quality evaluation on the generated class-conditioned data, (2) quality evaluation on the generated subject-conditioned data, and (3) how the generated data could help classification performance.

**Summary Of The Review:**

I suggest rejection. The paper lacks methodology novelty, and the evaluation procedure lacks benchmarking and quantitative evaluations.

---

### Official Review · Reviewer_fUSb · 2022-10-24

**Confidence:** 4
**Correctness:** 4
**Technical Novelty And Significance:** 2
**Empirical Novelty And Significance:** 2
**Recommendation:** 5

**Clarity, Quality, Novelty And Reproducibility:**

-The range of chromaticity bars in Figure 2 and Figure 5 is different. So some visual comparisons seem unfair.

-Why are some EEG-GANs not compared as baseline methods?

Please see the other specific comments in Strength And Weaknesses.

**Strength And Weaknesses:**

*Strength

-This work explored the application of the diffusion probabilistic model in EEG analysis.

-A large number of experiments were carried out on the manuscript.

*Weaknesses

- Technological innovation is limited. It is just a simple application of diffusion probabilistic model in specific fields.

-Although some evaluation indicators are effective. But I am worried about the value of practical application. As the authors mentioned, the classification performance on EEGNet cannot be improved.

-The details of the model are too simple. For example, what is Swish in the model diagram.


**Summary Of The Paper:**

The work shows a way of generating brain signals that can be useful in augmentation tasks. The current work shows that DPM-based brain signal generation is a very feasible task and can be used to create datasets that help improve deep learning models in classification tasks.


**Summary Of The Review:**

This is a typical application study. The diffusion probabilistic is simply applied to EEG. Some details still need to be clarified by the authors.

---

### Official Review · Reviewer_e92U · 2022-10-26

**Confidence:** 4
**Correctness:** 2
**Technical Novelty And Significance:** 2
**Empirical Novelty And Significance:** 2
**Recommendation:** 3

**Clarity, Quality, Novelty And Reproducibility:**

Clarity: Overall, the paper is generally readable, well-organised and the relevant authors (DPM, distillation) cited. However, several sections, particularly methods and results are confusing. Figure captions should be more informative, axes labeled, legends and acronyms need to be defined in the tables/figure captions.

Quality: The most relevant set of results to support the authors claims on data augmentation, section 5.4, is the weakest section of the paper. Statistical analyses are needed to evaluate the significance of the results.

Novelty: The authors apply recent methods in the literature (DPM, progressive distillation) to EEG signal generation. Authors claim this is the first application of DPM to EEG data.

Reproducibility: Additional details need to be described to better reproduce the findings.

**Strength And Weaknesses:**


Strengths:

- Authors present qualitative and quantitative results on the generated EEG signals, generating user-specific data, the impact of data augmentation on the downstream task (EEG signal classification) and progressive distillation to improve inference time.

- Results presented from two BCI applications (event related potential, motor imagery).

Weaknesses:

-    The results that are most relevant to data augmentation (section 5.4) do not really support the authors claims. "These observations
imply that although EEGWave was able to learn the main characteristics of the real signals, it could not produce signals that are diverse enough to regularize EEGNet."

- Figure and table captions need to be more informative to better interpret presented results. Are they subject-specific or presented in aggregate? Within-subject performance is more relevant in BCI applications as trends may sometimes be linked to EEG signal quality (signal-to-noise ratio). If presenting results in aggregate, an analysis comparing within-subject performance across all conditions needs to be included.

-    Since the authors contrasts DPM with RWGAN, classification results (not just assessment of the generated EEG signals) with synthetic data generated with GAN need to be included given that DPMs are slower (inference time also needs to be included for comparison). This is relevant in assessing the trade-off. Signal fidelity may not necessarily be as relevant if it has minimal/no impact on the performance of the classification task (the model only learns the most relevant features to distinguish between classes).




**Summary Of The Paper:**

This paper applies recent generative model, diffusion probabilistic model (DPM), to generate synthetic electroencephalography (EEG) data with utility demonstrated in two BCI applications (event-related potentials and motor imagery) using publicly available BCI datasets.

**Summary Of The Review:**

Recommendation is based mainly on the weaknesses outlined above.

Other comments
-	“Finding publicly available brain signal datasets that meet all requirements is a challenge.”
         o	What do authors mean by “all requirements”?

-	“However, the spatial resolution of the technology is quite poor due to its heavy dependence on the number of electrodes used for signal recording and non-invasiveness.”
         o	Authors should expand on relevance of this to the proposed work.

-	“The size and quality of publicly available data sets are limited, also often insufficient and imbalanced.”
         o	There are several publicly available BCI datasets. Authors should provide evidence (references, characteristics of publicly available datasets, etc.) to support this claim.
         o	The traditional approach to training machine learning models for BCI use is to rely on user-specific data. How are publicly available datasets relevant/useful within this training paradigm?

-	“We chose only the RWGAN (Panwar et al. (2020)) as a baseline model because other deep learning models in the literature were either incompletely documented, designed only for single-channel EEG generation, or did not work based on the given information in the published work.”
          o	Authors should reference the other models that were considered.
          o    What do authors mean by “did not work”? Needs to be more specific.

-	Diederik P Kingma, Tim Salimans, Ben Poole, and Jonathan Ho. On density estimation with diffusion models.
         o	Update title of paper in reference

-	More informative caption for Figure 1.

-	What are EEGWave, EEGWave 1x, EEGWave 1024 step, EEGWave 1 step? Figure 1 caption needs to be more description/informative.

-	“We give both qualitative and quantitative results”
         o	Distinguish between the qualitative and quantitative results listed, and include a brief description of each measure.

- The value of the subject-specific result section (section 5.3), especially to the classification task is not clear. One limitation is that the model is restricted to the subjects in the training dataset (subject labels are one-hot encoded), so needs to be retrained each time a new subject is added. Based on section 5.4, data from several subjects are combined to create the training/validation sets.

-	Are results presented in the figures/tables from one subject? Mean aggregate over all subjects? This needs to be specified in the captions. Preferably, subject-specific results are desirable, especially with relatively few subjects (18 and 9). Waveforms can be illustrated with data from a few subjects. If presented in aggregate, what are the number of subjects and the standard deviation for the measures presented in the table? What does bold mean in each table?

-	What do labels 0, 1, 2, 3 in Figures 6 and 7 mean. Label y-axes in Figures 8 and 9. What is "Base" in figures 8 and 9?

-	“Although the mixed training gave slightly better accuracy on the validation subset during training, it did not improve the accuracy of the test set. On the contrary, with the pre-training approach, the model converged faster and achieved better accuracy on the test.”
      o    Reference the relevant figures in the text. Are these aggregate results?

-	“The number of training iterations in steps 2 and 1 is doubled, following the work of _.”

-	“We believe the current work shows that DPM-based brain signal generation is a very feasible task and can be used to create datasets that help improve deep learning models in classification tasks.”

-   “The metrics commonly used in image synthesis tasks often give contrary results in brain signal generation.”
       o  Can authors explain what this means, especially "contrary results"? How is this relevant in interpreting results. What measures do the authors propose that are more relevant to brain signal generation?

---

### Official Review · Reviewer_bBMm · 2022-10-30

**Confidence:** 3
**Correctness:** 3
**Technical Novelty And Significance:** 2
**Empirical Novelty And Significance:** 2
**Recommendation:** 5

**Clarity, Quality, Novelty And Reproducibility:**

Clear, but not terribly original demonstration that diffusion models have a potential for EEG signal generation.

**Strength And Weaknesses:**

# Strengths

-   The authors have shown that diffusion probabilistic models (DPMs) are promising for generating brain data.
-   The problem to be solved is sufficiently substantiated in the introduction.
-   Two different datasets with different tasks are considered

# Weaknesses

-   It is not clear how the data was divided into train and test sets in section 5.2
-   a more detailed analysis of the specificity of subjects is missing (Section 5.3):

    1.  it would be good to supplement with a comparison

    of the specificity of subjects with other methods,

    1.  It is interesting to see the class of problems where such a property of the method is an advantage, and in which it is a limitation. Does this mean that this method cannot generate new subject data?
-   In section 5.4, the conclusion that pretraining on the generated data gives a gain is not obvious from the form of the matrix. Perhaps a general statistic would help.

DDIM - no decryption of the abbreviation One-hot - typo The number of training iterations in steps 2 and 1 is doubled, following the work of . - link missing

**Summary Of The Paper:**

The authors proposed the use of diffusion probabilistic models (DPMs) to generate EEG data in order to increase the size of datasets for training brain-computer interface classifiers. The qualitative and quantitative characteristics of the generated dataset were compared to the RWGAN model. The DPM model turned out to be comparable with RWGAN. The dataset of evoked potentials and the task of separating imaginary movements were taken as initial datasets. The authors note that DPM is able to generate EEG recordings that preserve the individual characteristics of subjects. The possibility of increasing the performance of EEGNet on the BCI4D2a dataset (classification of imaginary movements) was considered by expanding the dataset with generated data and by pretraining the dataset on these data. The second approach showed better results.

**Summary Of The Review:**

A paper applying diffusion models to the task of EEG data generation. Although the idea is interesting and the results are potentially promising, as mostly an application, the paper lacks in rigorous evaluation and empirical demonstration.

---

### Decision · Program_Chairs · 2023-01-20

**Decision:**

Reject

**Justification For Why Not Higher Score:**

Novelty and technical contribution is limited.

**Justification For Why Not Lower Score:**

N/A

**Metareview: Summary, Strengths And Weaknesses:**

Summary: The authors proposed the use of diffusion probabilistic models (DPMs) to generate EEG data in order to increase the size of datasets for training brain-computer interface classifiers.  The possibility of increasing the performance of EEGNet on the BCI4D2a dataset (classification of imaginary movements) was considered by expanding the dataset with generated data and by pretraining the dataset on these data.

Strengths: The authors have shown that diffusion probabilistic models (DPMs) are promising for generating brain data. Two different datasets with many tasks are considered, including class-conditioned generation, subject-conditioned generation, and classification.

Weakness: Technological innovation and novelty is limited. It is a simple application of diffusion probabilistic model in specific fields. Statistical analyses are needed to evaluate the significance of the results. Some details still need to be clarified by the authors.